# Development of a digital, self-guided return-to-work toolkit for stroke survivors and employers using intervention mapping

Kristelle Craven [1,2]*, Jain Holmes[1,2], Jade Kettlewell[1,2,3], Kathryn Radford[1,2]

1 Centre for Rehabilitation & Ageing Research, Injury, Recovery and Inflammation Sciences, School of Medicine, University of Nottingham, Nottingham, United Kingdom, 2 National Institute for Health and Care Research Nottingham Biomedical Research Centre, Nottingham University Hospitals National Health Service Trust, Queen's Medical Centre, Nottingham, United Kingdom, 3 Centre for Academic Primary Care, School of Medicine, University of Nottingham, United Kingdom

* Kristelle.Craven1@nottingham.ac.uk

## Abstract

Stroke incidence is rising among working-age adults in high-income countries. Employers often lack knowledge and skills to support return-to-work post-stroke. In the United Kingdom, nearly 40% of stroke survivors stop working. Vocational rehabilitation is rarely accessible, and self-guided resources often lack tools to support practical application. This study developed a self-guided return-to-work toolkit for stroke survivors and employers. Steps 1–4 of the six-step Intervention Mapping approach were followed. Intervention goal, content, and design were informed by three online workshops with employers (n = 12) and meetings with an advisory group (n = 20), including stroke charity and trade union representatives, stroke survivors, healthcare professionals, and experts in human resources and vocational rehabilitation. Theory-based pretesting (task-based usability review, advisory group discussions) was shaped by prototype review with advisory group members, including employers (n = 4), stroke survivors (n = 7), and healthcare professionals (n = 4). Framework analysis was used to structure feedback related to acceptability, ease of use/learnability, accessibility, inclusivity, perceived usefulness, and technical or environmental issues. No personal data were analysed. The toolkit aims to empower stroke survivors and employers to plan and manage a sustainable return-to-work post-stroke. It exists as two Xerte eLearning packages, with accessibility features such as screen reader compatibility and keyboard navigation. The toolkit contains theory- and evidence-based content for a) stroke survivors and b) employers, and includes downloadable PDF tools. Stroke survivor-focused content provides guidance on identifying and disclosing support needs to employers. Employer-focused content guides employers in increasing and maintaining understanding of stroke survivors' work abilities, and implementing and monitoring tailored reasonable adjustments. Pretesting indicated the toolkit is comprehensive, empowering, and fosters open

**Data availability statement:** Access to anonymised workshop transcripts is restricted due to the potential inclusion of identifiable and confidential participant information. This data management approach was approved by the University of Nottingham Faculty of Medicine & Health Sciences Research Ethics Committee (Ref: FMHS 166-1122). Participants consented to the use of anonymised quotes in publications and to the sharing of anonymised data with other researchers. Full workshop transcripts may be shared upon request with qualified researchers for research purposes, following publication, and subject to research ethics approval and a data-sharing agreement. Requests should be directed to the data custodian, Professor Kathryn Radford (mczkar1@exmail.nottingham.ac.uk), or to the Research Ethics Committee at fmhs-researchethics@nottingham.ac.uk. Other study materials, including workshop PowerPoint slides, can be found at: http://doi.org/10.17639/nott.7520.

**Funding:** This work was supported by the Ossie Newell Foundation (no grant number; awarded to KC), the National Institute for Health and Care Research Applied Research Collaboration East Midlands (no grant number; awarded to KC, https://arc-em.nihr.ac.uk/), and the Elizabeth Casson Trust (individual award to KC, grant number not applicable, https://elizabethcasson.org.uk/). The funders had no role in study design, data collection and analysis, decision to publish, or preparation of the manuscript. The views expressed are those of the authors and not necessarily those of the funders, or the Department of Health and Social Care.

**Competing interests:** The authors have declared that no competing interests exist.

communication, offering key information and practical tools. Minor refinements and technical improvements were suggested. This toolkit addresses a gap in return-to-work guidance in the United Kingdom. Refinement, testing, and evaluation in real-world settings are needed.

---

## Author summary

More working-age people are having strokes, and many struggle to return to work afterward. In the United Kingdom, nearly 40% of stroke survivors stop working altogether. Employers often don't know how to help, and support services are limited or hard to access. In this project, we worked closely with stroke survivors, employers, healthcare professionals, and other experts to design a toolkit to guide employees and employers in planning a stroke survivor's return to work. We created two interactive online learning packages—one for stroke survivors and one for employers. Each offers practical advice and downloadable tools, such as how to talk about support needs, or make appropriate changes at work. We asked a group of stroke survivors, employers, and healthcare professionals to look at the toolkit and give feedback. The group reported that the toolkit content was clear, practical, and helpful, and that it encouraged open and honest communication. They suggested a few small changes and technical improvements. We believe this toolkit could make a difference for many people attempting to return to work after a stroke. Our next step is to make the suggested changes, and test how well it works in real-world settings.

## 1. Introduction

Stroke occurs when blood supply to the brain is restricted [1], causing brain injury, disabilities, or even death [1,2]. In the United Kingdom (UK), stroke incidence among adults aged <55 years increased by 67% between 2002 and 2018 (N = 94,567) [3]. Stroke has been linked to a wider range of disabilities than any other health condition [4], including impairments in vision, sensation, hearing, communication, emotion regulation, movement, and cognition [3]. Stroke survivors may also experience grief over identity loss, frustration, anxiety, and depression [5]. The effects of stroke can impair capability in all domains of daily life, including work [6].

Stroke survivors often experience return-to-work (RTW) challenges. For example, stroke survivors and employers have reported lacking understanding of stroke [7], and their RTW roles and responsibilities [8], unrealistic job demands, and unsupportive work climates [9–11]. Strong evidence shows that employer support, e.g., from managers and supervisors, is key for ensuring a sustainable RTW [12]. However, employers do not always have knowledge or guidance for communicating with and supporting stroke survivors to RTW [13].

UK national clinical guidelines recommend collaboration between stroke survivors, employers, and healthcare professionals [14,15]. For example, professionals who provide rehabilitative support for retaining-, or returning to and remaining in work (otherwise known as 'vocational rehabilitation' [VR]) [16]. However, VR provision in the National Health Service (NHS) is inconsistently available [17], due to funding gaps [18–20], lengthy waiting lists [21], and lack of referrals or employer engagement [20,22]. Stroke survivors and employers may access VR privately via their organisations (e.g., through employee assistance programs, occupational health services), or through UK government schemes, or the third sector. However, support offered may be inadequate because it is not stroke-specific, nor part of a comprehensive rehabilitative program [21]. Moreover, small- or medium-sized organisations (≤250 employees) may not perceive demand for, nor have resources to privately fund VR [23–25]. Simultaneously, there are national shortages of healthcare professionals with VR expertise, such as occupational therapists [26] and occupational health advisors [27]. In a 2018 UK survey (N = 11,134), 37% of working-age stroke survivors stopped working following stroke [28]. Reduction or loss of employment among UK stroke survivors cost £1.6 billion in 2015, and is predicted to rise 136% by 2035 [29].

In a randomised controlled trial (N = 583), stroke survivors of older age, or with greater post-stroke impairment, benefitted most from therapist-delivered stroke-specific VR (interaction p = 0.023 and p = 0.096, respectively) [30]. The authors suggested that younger stroke survivors with mild-to-moderate stroke severity may be capable of self-navigating and advocating for their RTW. Notably, only 119 (40.3%) of the intervention arm (n = 309) consented to therapist contact with their employer, and 67 (22.7%) were self-employed or had no employer. Thus, intensive therapist-delivered VR may not always be desired, and/or needed by this sub-group. According to the NHS Model for Stroke VR, level 3 services for those with less complex needs should include work-related information and advice, and signposting for further support [31].

In recent years, NHS and governmental reforms to improve sustainability through digital technologies have been introduced [32–34]. A potential benefit includes increased service user self-management of health, and enhanced access to and efficiency of VR services. Web-based guides to support stroke survivors and employers through the RTW process exist [35–38], but it is unclear how they were developed and/or evaluated. Cognitive development theories emphasise the crucial role that tools play in facilitating learning [39,40], yet sets of tools are rarely included within resources. Given the lack of available resources and inconsistent VR provision, stroke survivors and employers may benefit from a RTW toolkit intervention that is 'self-guided', i.e., an intervention without individual support from a therapist or trained person [41]. In this context, a toolkit refers to a collection of resources or 'tools' with a specific focus on a single audience [42]. Toolkits might include educational materials, assessment tools, and planning tools, to aid translation of evidence into practice [42,43]. Previous searches during a systematic review [13] and advisory group consultation did not identify any such interventions relating to stroke. However, self-guided RTW interventions for people with other conditions, e.g., cancer, mental illness, and/or their employers were identified [44–49]. Most had not been tested for effectiveness, but were acceptable and useful to employers [47,49] and employees with depression [50] or cancer [45]. In two trials investigating effectiveness, groups provided with web-based, self-guided RTW interventions reported statistically significant shorter median duration (in days) until first or full RTW [46,48]. One intervention was considered effective because it encouraged women recovering from gynaecological surgery to actively engage in their recovery [48].

This study aimed to develop a digital, self-guided RTW toolkit for stroke survivors and employers using the intervention mapping (IM) approach. IM is a well-established planning framework that guides multi-level development, implementation, and evaluation of complex interventions [51]. It incorporates community-based participatory research methods to ensure interventions match users' priority needs and contexts. This paper describes step-by-step how the Toolkit for Transitioning to Employment After stroke through Mutual support (TTEAM) was designed.

## 2. Materials and methods

### 2.1. Ethics statement

Ethical approval was obtained from the University of Nottingham Faculty of Medicine & Health Sciences Research Ethics Committee (ref: FMHS 166–1122). Formal, informed consent was obtained verbally from employer workshop participants.

The reporting of this study was guided by the GUIDance for rEporting intervention Development studies in health research (GUIDED) checklist [52] (S1 Table), and the Template for Intervention Description and Replication (TIDieR) checklist [53] (S2 Table). As this study focused on intervention development using IM, stakeholder input was summarised narratively to inform decision-making, rather than reported as verbatim excerpts.

### 2.2. TTEAM development process

Tasks completed during each IM step informed the work completed in the subsequent step, resulting in an intervention based on theory, empirical evidence, and practical information [54]. In this study, IM steps 1–4 were followed to develop TTEAM, with plans to complete refinement and steps 5–6 in future work. Descriptions of IM terminology are provided in Table 1. The IM process [51] is summarised in Table 2, alongside related study activities and outputs. A depiction of this process is shown in Fig 1. Detailed description of methods per IM step is provided in S1 Text.

### 2.3. Expert advisory group involvement

The IM approach recommends setting up a group of representatives from the target population/s, implementers, and other relevant stakeholders to be involved throughout, to ensure the intervention addresses community needs identified [51]. In July 2022 KC identified potential members via convenience sampling, and invited them to attend three advisory group meetings over two years, to advise on data collection (relating to the needs assessment and workshops), a 'logic model of change,' and pretesting of TTEAM prototypes. Potential members were selected based on their clinical, professional, and/ or academic expertise, or lived experience of stroke, VR, implementation science, human resources (HR), or occupational

**Table 1. Key terms from the IM approach.**

| Key term | Description |
|---|---|
| Logic model of the problem | A logic model depicting pathways of problem causation, i.e., how factors can influence or cause a health problem. |
| Behavioural outcomes | Overall behaviours to be performed by intervention users, as a result of taking part in the intervention. |
| Performance objectives | Sub-behaviours/actions that are needed to produce the behavioural outcomes. |
| Personal determinants | Factors internal to intervention users that influence the behaviours contributing to the health problem. Examples include cognitive factors, e.g., knowledge, attitudes, beliefs, self-efficacy, and capabilities, e.g., skills. |
| Change objectives | Change objectives state what the intervention needs to change in order to facilitate achievement of performance objectives (and subsequently, the behavioural outcome/s). |
| Matrices of change | For each behavioural outcome, a matrix of change is constructed. This is a table with performance objectives in the left-hand column, and determinants across the top row. Each cell of the matrix is then judged to see whether a determinant will influence achievement of the corresponding performance objective. Change objectives are then defined to theoretically remove any negative influence of the determinant and facilitate achievement of the performance objective. |
| Logic model of change | A logic model depicting theorised pathways of intervention effects, i.e., what will change as a result of the intervention being implemented. |
| Behaviour change methods (otherwise known as 'intervention' methods) | General theory- and evidence-based techniques for influencing determinants of behaviours and environmental conditions. |
| Practical applications | The ways in which intervention methods are delivered, e.g., a video showing an ideal behaviour in a given scenario. These should fit with the intervention context and user group/s. |

**Table 2. The six-step IM process.**

| IM step and recommended tasks | Research study activities | Outputs |
|---|---|---|
| **Step 1. Logic model of the problem** | | |
| • Establish and work with planning group | Expert advisory group set up | Advisory group available to advise on project delivery and intervention development |
| • Conduct needs assessment to create logic model of the problem | Mixed methods synthesis of employers' needs when supporting stroke survivors to RTW post-stroke (qualitative review [25 studies], survey [n = 50], and interviews [n = 7]) [13,55] Workshop 1 (n = 7) | Logic model of the problem |
| • Describe intervention context (including setting, community, population) | Needs assessment and consultation with advisory group | Description regarding *who* needs *what* from TTEAM |
| • State intervention goals | Workshop 1 (n = 7) | Decision made on TTEAM goal and priority areas. Logic model of the problem refined. |
| **Step 2. Intervention outcomes and objectives: Logic model of change** | | |
| • State behavioural and environmental outcomes <br> • Specify performance objectives to achieve the outcomes <br> • Select determinants for the outcomes | All outcomes and objectives constructed by KC (based on logic model of the problem, and group discussion during Workshop 1 [n = 7]) | Logic model of change |
| • Create matrices of change objective <br> • Create logic model of change | Feedback on logic model of change obtained from research team and expert advisory group | |
| **Step 3. Intervention design** | | |
| • Decide on intervention scope, sequence, themes, and components <br> • Select evidence- and theory-based intervention methods <br> • Decide on practical applications to deliver intervention change methods | Feedback and ideas obtained from research team and Workshop 2 participants (n = 5) | TTEAM plan (including scope, sequence, and components) |
| **Step 4. Intervention production** | | |
| • Plan intervention materials <br> • Draft key messages, approaches, and intervention materials | Feedback on TTEAM plan, materials, and key messages obtained from Workshop 3 participants (n = 4) | Refined TTEAM key messages and materials. Initial intervention prototypes created on Xerte (one for stroke survivors, one for employers) |
| • Pre-test, refine, and produce intervention materials | Consultation with expert advisory group | Feedback to inform refinement of TTEAM prototypes |
| **Steps 5 (Intervention implementation plan) and 6 (Evaluation plan) to be constructed in future work.** | | |

health (OH). Informed consent was not required, as their role was to advise on the project and oversee its progress, rather than contribute personal data [56]. For example, during the 'pre-testing' of TTEAM (IM step 4), they were invited to provide expert and practical feedback on prototypes in a consultative capacity, and not as research participants. Although some informally disclosed their stroke-related impairments to KC (first author), no personal data were collected.

Over a two-week period (September 2024) advisory group members accessed and reviewed the TTEAM prototype(s) of their choice. During this time, they were asked to complete a series of guided tasks simulating typical use, such as adjusting accessibility settings, navigating pages, viewing content in various formats (text, video, PDF), and exploring the prototype(s) more broadly.

Group members provided feedback via email or through participation in a Microsoft Teams group discussion. Feedback questions were based on constructs from the Technology Acceptance Model [57,58], System Usability Scale [59], and International Classification of Functioning, Disability and Health [60] (S3 Table), including acceptability, ease of use/learnability, accessibility and inclusivity, perceived usefulness, and issues affecting its use (technical or environmental). Framework analysis [61] was used to structure consultative feedback from advisory group meeting transcripts and email

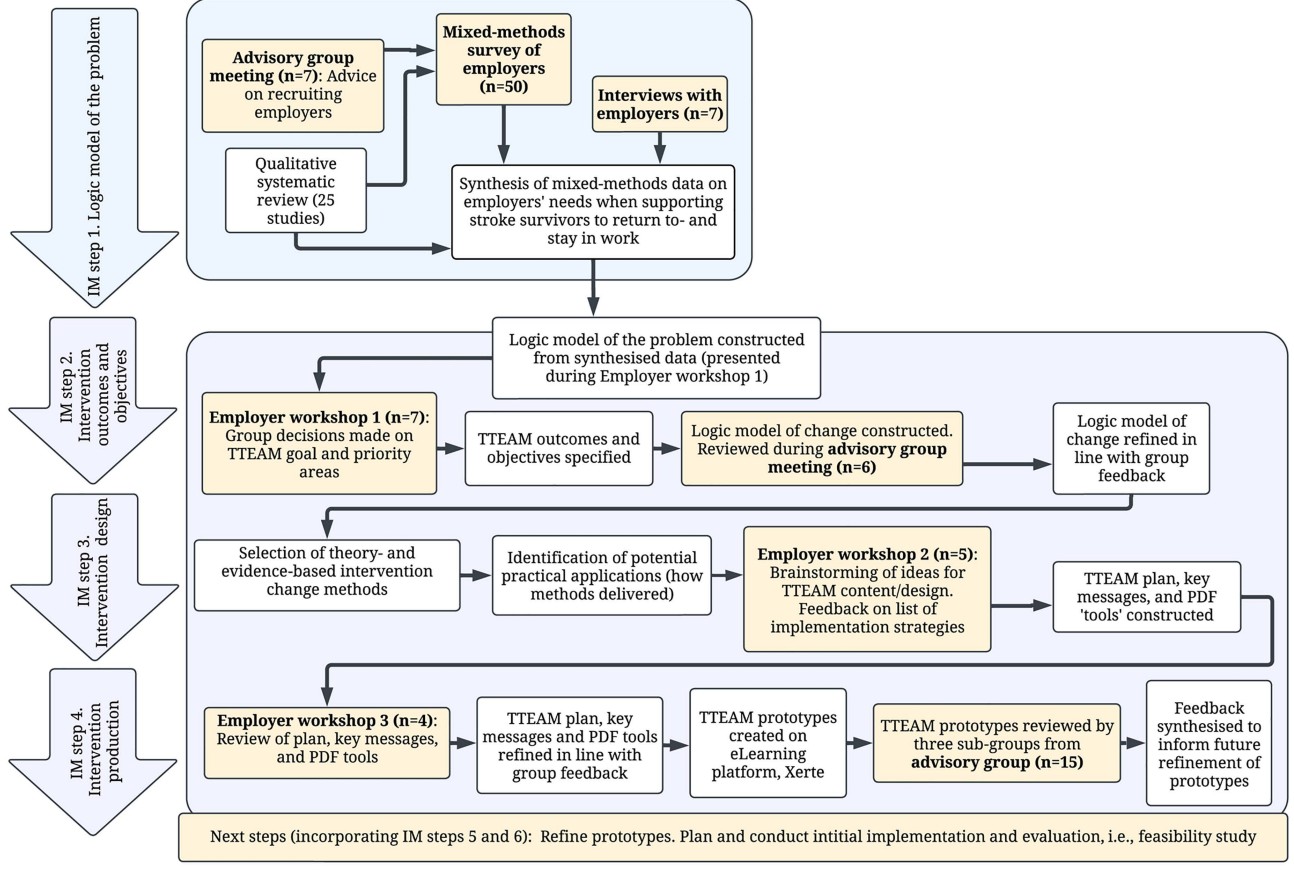

**Fig 1. Flow diagram of the TTEAM development process, with IM step 1 fully reported elsewhere [13,55].**

responses, guided by the question constructs. Feedback was organised using NVivo version 12 [62] and Microsoft Excel (version 16.65). KC (first author) had experience of framework analysis and independently carried out the process. Coding decisions and interpretations were discussed with the research team to enhance credibility of the feedback summary.

## 2.4. Workshops with employer participants

Aside from the advisory group activities, three workshops were conducted with employer participants to facilitate decision-making on TTEAM's priority areas and goals (Workshop 1; August 2023), design and content (Workshop 2; October 2023), and to obtain feedback on the plan, materials, and key messages for TTEAM (Workshop 3; February 2024).

Employers were recruited through convenience and snowball sampling. The study was advertised February 2023-January 2024, through social media, gatekeepers, and business networking events in the Midlands, UK. Eligible participants were aged 18 years or older; worked in an occupational role involving staff responsibility; and proficient in use of English language. Potential participants contacted KC via email to receive participant information. Sample sizes for the workshops were determined pragmatically, based on participant availability and engagement in the co-design process, in line with the iterative and participatory nature of the IM approach [54].

In total, three workshops (of two hours duration each) were conducted via Microsoft Teams. Participation involved attending one or more of these workshops. Verbal informed consent was taken one-to-one prior to the workshop/s, and documented on a Microsoft Word form, signed on the participant's behalf. Completed verbal consent forms (.docx) were

sent to participants. Workshops were recorded and transcribed using Microsoft Teams. Recordings (.mp4), completed consent forms (docx), and anonymised transcriptions (.docx) were saved securely on Microsoft Teams.

## 3. Results

### 3.1. Description of the expert advisory group

Advisory group members (n = 20) included seven stroke survivors (two also representing a stroke charity), one stroke charity representative, six healthcare professionals experienced in VR (occupational therapists, occupational health and rehabilitation professionals, and a health psychologist), a trade union representative, two researchers with expertise in IM and implementation science, and two HR experts. An additional six members (three stroke survivors, two managers, and one business owner) were recruited during the pretesting stage (IM step 4), via a local stroke research partnership group and personal and professional networks of the research team.

### 3.2. Occupational roles, settings, and workshop attendance of employer participants

Participant characteristics (stroke survivor status, occupational role, organisational setting) and their attendance of workshop 1 (n = 7), workshop 2 (n = 5), and workshop 3 (n = 4) are included in Table 3. As noted in the table below, some were also stroke survivors and/or professionals working in healthcare or HR.

### 3.3. Step 1. Logic model of the problem

#### 3.3.1. Needs assessment: 'Who' needs 'what' from the intervention?. The needs assessment is reported fully elsewhere [13,55]. Findings showed employers with the following demographics would benefit *most* from TTEAM:

• Working in small or medium-sized organisations (≤250 employees)

• No HR or occupational health support

• No post-stroke RTW experience

**Table 3. Occupational roles, settings, and workshop attendance of employer participants.**

| Participant identification number | Occupational role/s | Stroke survivor (yes/no) | Organisational setting | Workshop/s attended |
|---|---|---|---|---|
| 03 | Clinical supervisor/ Occupational therapist | No | Neurorehabilitation service in National Health Service (NHS) | Workshop 1 |
| 09 | Plant manager | Yes | Construction organisation (size not reported) | Workshop 1 |
| 10 | Line manager | No | Large manufacturing organisation | Workshops 1, 2, and 3 |
| 11 | Clinical supervisor/ Occupational therapist | No | Neurorehabilitation service in NHS | Workshops 1 and 2 |
| 12 | Clinical supervisor/ Occupational therapist | No | Neurorehabilitation service in NHS | Workshop 1 |
| 13 | Occupational health advisor | No | Large manufacturing organisation | Workshop 1 |
| 14 | HR advisor | No | Stroke charity | Workshop 1 |
| 15 | Clinical supervisor/ Occupational therapist | No | University | Workshops 2 and 3 |
| 16 | Information Technology manager | Yes | Large manufacturing organisation | Workshop 2 |
| 17 | Administration manager | Yes | Stroke charity | Workshop 2 |
| 05 | Small business owner | Yes | Private rehabilitative therapy service | Workshop 3 |
| 20 | Occupational health consultant | No | NHS | Workshop 3 |

Additionally, other employers may benefit from the following:

- **Education about stroke:** Any employer without post-stroke RTW experience, and/or without support from HR/occupational health professionals.

- **Support with understanding the RTW process, including roles/responsibilities:** Employers in small or medium-sized organisations, without HR/occupational health support.

- **Support with improving perceived competency for carrying out supportive RTW actions** (e.g., finding out what to expect from the stroke survivor in their working role): Employers without post-stroke RTW experience.

Findings from the needs assessment informed the development of an initial logic model of the problem (S1 Fig).

  **3.3.2.  Refined logic model of the problem.**  Based on the needs assessment and selection of priority areas in Workshop 1, the logic model of the problem was refined (S2 Fig). Employers' lack of knowledge about stroke, organisational RTW policies and procedures, and roles/responsibilities meant they over- or under-estimated stroke survivors' abilities, and hesitated in making reasonable adjustments. Stroke survivors did not disclose their needs to employers, because they feared highlighting their limitations. Workshop 1 participants suggested this was due to pre-conceived ideas of how an employer would respond, and fear of dismissal and discrimination, as well as stroke survivors (and employers) not understanding roles of stakeholders, e.g., healthcare professionals, who could help them. Together, these behaviours and personal determinants meant that stroke survivors' workloads and RTW process durations were unsuitable, and/or they were not given timely supervisory support and reasonable adjustments.

During the first advisory group meeting, the consensus was that TTEAM should consist of a stroke survivor version, and an employer version. During Workshop 1, it was decided TTEAM's goal should be to empower employers and stroke survivors to plan and manage a timely and sustainable RTW post-stroke. Priority areas included: 1) Stroke survivors disclosing essential needs to their employer; 2) Employers increasing and maintaining understanding of stroke survivors' abilities; and 3) Employers providing reasonable adjustments for stroke survivors when needed.

## 3.4.  Step 2. Logic model of change

In IM, the logic model of change depicts pathways of intervention effects, i.e., how the intervention leads to desired outcomes [54]. Following presentation of the model and discussion at the second advisory group meeting, additional determinants relating to stroke survivors included: not knowing they are ready to plan and prepare for a RTW, believing work caused their stroke, or worrying about future impacts of work on their health. The Theoretical Domains Framework (TDF) is an integrative framework based on 84 behaviour change theoretical constructs [63], relevant to constructs included within the IM taxonomy of intervention methods [54,64]. Using the TDF, stroke survivor determinants were mapped onto the domains of behavioural regulation, beliefs about capabilities, beliefs about consequences, emotion, knowledge and intentions, i.e., conscious decision to perform a behaviour.

Employer determinants were mapped onto the domains of emotion, beliefs about capabilities, beliefs about consequences, knowledge, skills, and professional role and identity. The matrices of change are presented in S4 Table. Fig 2 shows the refined logic model of change for TTEAM.

## 3.5.  Step 3. Design of TTEAM

  **3.5.1.  Selection of theory- and evidence-based behaviour change methods.**  Feedback on TTEAM's scope, sequence, and components was obtained during Workshop 2, and are outlined in the remainder of this sub-section. IM's taxonomy of behaviour change [64] indicated various relevant theoretical behaviour change methods for the change and performance objectives (shown in S5 Table, alongside initial ideas on practical applications). Persuasive communication; arguments; participation; public commitment; modelling; planning coping responses; implementation intentions; goal

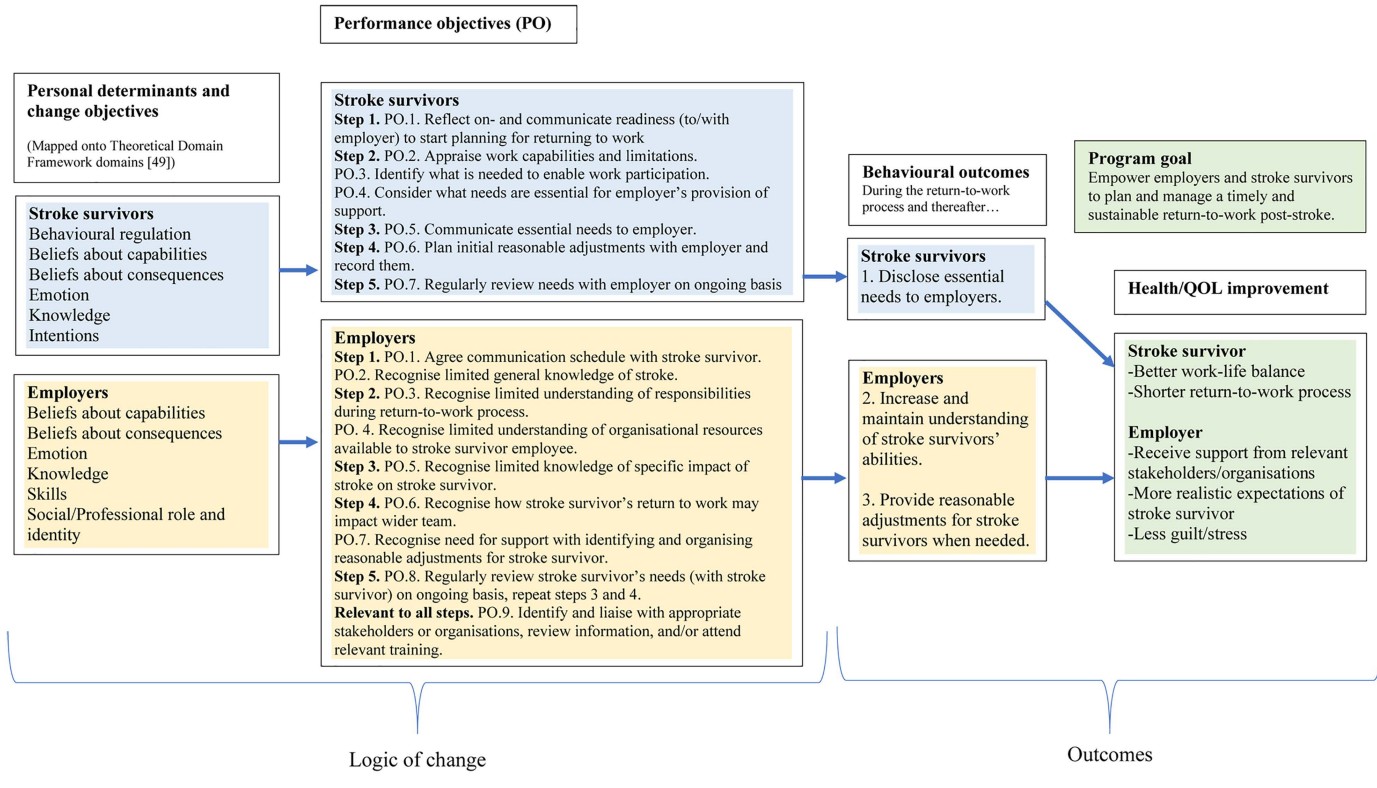

**Fig 2. Refined logic model of change for the TTEAM intervention.**

setting; active learning; advance organisers; chunking; self-monitoring of behaviour; and environmental re-evaluation were change methods identified for both stroke and employer versions. Information about others' approval and mobilising social support were specifically identified for the employer intervention.

Definitions of the selected change methods, underlying theories, and linked TTEAM steps and performance objectives are shown in Table 4.

**3.5.2. Selection of practical applications.** Behaviour change methods require feasible and suitable practical application to be effective. KC combined initial ideas for practical applications with those suggested during the first advisory group meeting and Workshop 2 (S6 Table). In line with IM guidance [54], practical applications were refined according to feasibility.

Workshop 2 participants suggested a glossary of potential stakeholders' roles, as well as contact details for relevant organisations offering advice, information, and support. TTEAM tasks recommend seeking support when needed, e.g., employers to learn specific impact/s of stroke on an employee's work capability. Videos, text, and images were also suggested and included for conveying key messages. Within TTEAM for example, videos with stroke survivors convey personal stories whilst relaying key messages, e.g., sustainable RTW post-stroke is possible. Advice from an HR expert is also included, e.g., regarding the importance of regular, ongoing reviews of reasonable adjustments.

In the stroke survivor version, key messages relate to causes and effects of stroke, stroke recovery (as being individualistic and potentially long-term), and the importance of early, regular communication with employers. Also included are legal definitions of disability and reasonable adjustments, and information about work simulation tasks and workplace buddies.

**Table 4. Definitions of selected behaviour change methods and their underlying theory, linked to TTEAM steps and performance objectives.**

| Behaviour change method | Definition (as taken from the IM approach's taxonomy of behaviour change methods) [64] | Underlying theory | Stroke survivor version: TTEAM step and related performance objective/s (PO) (See Fig 2 for PO definitions) | Employer version: TTEAM step and related performance objective/s (PO) (See Fig 2 for PO definitions) |
|---|---|---|---|---|
| Persuasive communication | Guiding individuals and environmental agents toward adoption of an idea, attitude, or action by using argument or other means. | Communication-Persuasion Matrix [65]; Elaboration Likelihood Model [66,67]; Diffusion of Innovations theory [68] | 1 (PO.1)<br>2 (PO.2)<br>5 (PO.7) | All steps and POs |
| Arguments | Using a set of one or more meaningful premises and a conclusion. | Communication-Persuasion Matrix [65]; Elaboration Likelihood Model [66,67] | 3 (PO.5) | 1 (PO.1) |
| Participation | Assuring engagement of participants in problem-solving, decision making, and change activities. | Diffusion of Innovations theory [68] | 4 (PO.6)<br>5 (PO.7) | 5 (PO.8) |
| Public commitment | Stimulating pledging to perform the desired behaviour, and announcing that decision to others. | Theories of Automatic, Impulsive and Habitual Behaviour [69,70] | 4 (PO.6)<br>5 (PO.7) | 5 (PO.8) |
| Modelling | Providing an appropriate model; being reinforced for their desired action. | Social Cognitive Theory [71,72]; Theories of Learning [73] | 1 (PO.1)<br>2 (PO.2)<br>3 (PO.5) | 1 (PO.1)<br>4 (PO.6, PO.7) |
| Planning coping responses | Prompting participants to list potential barriers. | Attribution Theory [74]; Theories of Goal-Directed Behaviour [75] | 4 (PO.6) | 1 (PO.1)<br>4 (PO.6) |
| Implementation intentions | Prompting if-then plans, linking situational cues with responses effective in attaining desired outcomes. | Theories of Goal-Directed Behaviour [75]; Theories of Automatic, Impulsive and Habitual Behaviour [69,70] | 5 (PO.7) | Not selected |
| Goal setting | Prompting planning what the person will do, including a definition of goal-directed behaviours that result in the target behaviour. | Goal-Setting Theory [76]; Theories of Self-Regulation [77] | 5 (PO.7) | 1 (PO.1)<br>2 (PO.3)<br>4 (PO.7) |
| Active learning | Encouraging learning from goal-driven and activity-based experience. | Elaboration Likelihood Model [66,67]; Social Cognitive Theory [71,72] | 1 (PO.1)<br>2 (PO.2, PO.3, PO.4)<br>3 (PO.5)<br>4 (PO.6)<br>5 (PO.7) | 1 (PO.1, PO.2)<br>2 (PO.3, PO.4)<br>3 (PO.5)<br>4 (PO.6, PO.7)<br>5 (PO.8) |
| Advance organisers | Presenting an overview of the material than enables a learner to activate relevant schemas so that new material can be associated. | Theories of Information Processing [78–81] | All steps and POs | All steps and POs |
| Chunking | Using stimulus patterns that may be made up of parts but that one perceives as a whole. | Theories of Information Processing [78–81] | Not selected | 2 (PO.3)<br>3 (PO.5)<br>4 (PO.6) |
| Self-monitoring of behaviour | Prompting the person to keep a record of specified behaviour/s. | Theories of Self-Regulation [77] | 5 (PO.7) | 1 (PO.1)<br>2 (PO.3)<br>5 (PO.8) |
| Environmental re-evaluation | Encouraging realising the negative impact of the unhealthy behaviour. | Transtheoretical Model [82] | 1 (PO.1)<br>3 (PO.5) | 5 (PO.8) |

In the employer version, key messages are identical with greater emphasis on stroke impacting individuals differently, employers' roles/responsibilities (including UK legal obligations), the importance of consent and confidentiality, and importance of assessing health and safety risks relating to the stroke survivor's RTW.

The key focus and remaining applications per TTEAM step are summarised in Table 5.

**Table 5. Key focus and applications per TTEAM step.**

| TTEAM step | Key focus and applications included | |
| --- | --- | --- |
| | Stroke survivors | Employers |
| 1 | **Readiness to start planning for a RTW** | **Stroke and communication** |
| | **Pros and cons activity, a readiness checklist, reflective prompts, and a template script to help stroke survivors reflect on and communicate readiness to plan a RTW.** | A five-minute quiz to informally assess stroke knowledge, interactive diagram to learn about stroke disabilities, **reflective prompts on confidence/skills for supporting a stroke survivor, communication script/schedule templates*, record of additional learning activities, and a tool to facilitate identification of potential communication issues, and coping strategies** (if stroke survivor has dysphasia for example). |
| 2 | **Appraisal of capabilities, limitations, and needs** | **Employers' roles and responsibilities** |
| | **Job demands analysis tool and self-monitoring diary to help them appraise their RTW needs.** | Drag-and-drop exercise regarding legal obligations and other responsibilities, CHEAP acronym to aid memory as to what makes an adjustment 'reasonable', reasonable adjustments crossword puzzle, **worksheet to aid identification and communication of organisational resources.** |
| 3 | **Disclosure of needs to employer** | **Understanding stroke survivors' capabilities, limitations, and needs** |
| | **Template script to aid disclosure of limitations and needs to their employer.** | Drag-and-drop activity to test skill in linking work performance difficulties to disabilities, CARES acronym to aid memory on communication style, **job analysis tool* (employer version to be completed with stroke survivor)**, 5-minute multiple choice quiz to assess how to gain information about stroke survivors. |
| 4 | **Planning and recording reasonable adjustments** | |
| | **Tools to support the planning and recording of reasonable adjustments* and RTW in general, e.g., a risk assessment template*, a tool to facilitate identification of potential problems upon RTW and coping responses, and templates for a reasonable adjustment passport*, wellbeing action plan*, and RTW plan*.** Employer version also has a multiple-choice quiz on planning and recording reasonable adjustments, drag-and-drop activity regarding reviewing health and safety risks, and RISE acronym to aid memory on how to introduce stroke survivor back into team. | |
| 5 | **Ongoing review of capabilities, limitations, needs and reasonable adjustments** | |
| | **Contract template for conducting regular reviews of the stroke survivor's needs and support*.** Both versions have instructions for setting up and conducting reviews. Employer version also has brief multiple-choice quiz on stroke recovery and what to include in reviews, and **reflective prompts on additional learning needs.** | |

Bold font indicates where a TTEAM activity is provided as a PDF tool/form.

*Indicates where it is recommended a stroke survivor and employer complete an activity together.

**3.5.3. Feedback on design features for TTEAM.** Workshop 2 participants suggested TTEAM be visually stimulating, with different pathways for stroke survivors and employers, and additional resources as downloadable PDFs. Participants thought online (website or eLearning), lifetime access to TTEAM should be given immediately following the stroke survivor's discharge from hospital, and that TTEAM steps should be quick and easy to complete, with content in audio format also.

### 3.6. Step 4. TTEAM intervention production

**3.6.1. Workshop 3 feedback on the intervention plan.** Several participants were unable to attend Workshop 3, so it was held across two sessions to maximise input. The first session was attended by a line manager and an OH consultant, and focused on the plan for the employer TTEAM version. The second session was attended by two employers who were also stroke survivors (and therefore able to give a stroke survivor perspective also). This session focused on the plan for the stroke survivor version. Plans for both TTEAM versions are presented in S7 Table.

Changes were suggested to optimise the learning experience. During the first session, employers suggested providing brief facts (statistics) within information about stroke (TTEAM step 1), emphasising benefits of work for stroke recovery

(TTEAM step 1), and providing reminders about consent and confidentiality throughout TTEAM. It was also suggested the job demands analysis tool (TTEAM step 3) include consideration of functional tasks, and how each difficulty listed might impact the stroke survivor's work performance. In the list of reasonable adjustments provided (job demands analysis tool), hyperlinks were suggested to enable further learning. It was emphasised that reasonable adjustments be defined clearly, e.g., 'reasonable' meaning affordable, effective, practical and safe to implement. Additional suggestions included guidance on informing the wider team about the stroke, and information on work trial benefits, and stroke recovery, e.g., as potentially long-term and gradual. During the second session, additions to the stakeholder glossary and overview of organisations, improved accessibility of the self-monitoring diary (TTEAM step 2), and inclusion of example work tasks in the job demands analysis tool were suggested (TTEAM step 2).

**3.6.2. Design features of TTEAM.** The TTEAM prototypes are hosted on Xerte [83] as two eLearning packages: one for stroke survivors and one for employers. The final two TTEAM steps are identical across both versions, and contain activities to be completed jointly by stroke survivors and employers (Table 5). These activities enable tailoring of support to stroke survivors' needs, e.g., through identification and recording of suitable reasonable adjustments via the passport tool provided. Reflective activities are also included to guide individual users in decision-making, and seeking additional information and support where needed (Table 5). Xerte is an open-source content creation platform designed to enable non-technical authors to rapidly create accessible, interactive, and engaging learning resources [83]. Xerte itself is free-to-use, and the University of Nottingham hosts TTEAM on a free, unlimited basis. Each prototype contains a pop-up menu to navigate to different sections. PDF 'tools' are included to support completion of suggested tasks. Interactive elements were included only in the employer version to ensure stroke survivors with restricting impairments, e.g., hand-eye coordination challenges, were not excluded or limited in their use of TTEAM. Users may utilise TTEAM on unlimited, self-guided basis via a URL link (with or without password access). Each TTEAM step is designed to take 15 minutes to complete (total duration: 1 hour, 15 minutes), with additional time needed for optional tasks.

**3.6.3. Pretesting activity: feedback from the expert advisory group.** Three meetings were held separately with stroke survivors (n = 7), employers (n = 4), and occupational therapists (n = 2). A trade union representative and an OH expert provided email feedback. One stroke survivor reviewed TTEAM using a mobile phone; all others used a laptop or desktop computer. One stroke survivor also used a tablet. The stroke survivor version of TTEAM was reviewed by all seven stroke survivors, one employer, one trade union representative, and one OH professional. The employer version was reviewed by four stroke survivors, all four employers, and two occupational therapists. Detailed summaries of their feedback are provided in S8 Table.

Overall, TTEAM was seen as comprehensive, useful, and empowering, offering key information and facilitating open communication between stroke survivors, employers, and the wider support team. Some expressed interest in using TTEAM themselves, and praised its ease of navigation, and clear, concise, and interactive content. To improve ease of use and learnability, amendments to instructions for use and inclusion of the five-step TTEAM process overview were suggested. To improve accessibility and inclusivity for users with cognitive and/or fatigue issues, it was suggested that initial text density be reduced. Other suggestions included removal of pop-up descriptions for images/videos, and production of audio TTEAM versions. Minor suggestions were made to improve accuracy/relevancy of content and the user experience, e.g., including PDF tools as an appendix at the end of TTEAM. Technical issues such as slow video loading, PDF/narrator-speech tool incompatibility, and inability to save progress were noted. To increase awareness and promote access to TTEAM, members suggested signposting via healthcare and HR professionals, the UK's Department for Work and Pensions, charities, non-profit organisations, and employers. Awareness events, a network of users, and promotional video were also suggested.

## 4. Discussion

This study shows how IM was applied in the co-design of TTEAM, a self-guided digital RTW toolkit for stroke survivors and employers. Relevant stakeholders were involved throughout to prioritise user needs, and guide decisions on

TTEAM content, goal, and design. TTEAM consists of two interactive eLearning packages on Xerte, with theory- and evidence-based content designed to empower stroke survivors and employers to plan and manage a timely, sustainable post-stroke RTW. Initial feedback conveyed that TTEAM is a comprehensive, empowering RTW resource that fosters open communication, and offers key information and practical tools.

In the UK and elsewhere, clinical guidelines for stroke recommend that self-guided rehabilitation resources be provided alongside traditional rehabilitation [84,85]. However, VR provision is limited in the UK [18,19]. In the NHS, for example, VR services are rarely commissioned, with limited access, inadequate resources, and inconsistent referral practices [18–22]. In 2021, only 7.4% of post-acute services audited through the Sentinel Stroke National Audit Programme for England and Wales had commissioned a VR service [86]. Following a randomised controlled trial (N = 583) of a stroke-specific, therapist-delivered VR intervention, the authors concluded that younger stroke survivors with mild-to-moderate severity may be able to self-navigate and advocate for their RTW [30].

TTEAM is a self-guided, digital RTW toolkit intervention containing components recommended for VR programs in the UK National Clinical Guideline for Stroke [84], including assessment of potential RTW barriers and facilitators, and action planning for overcoming barriers. A systematic review published in 2023 identified 12 studies reporting on VR interventions for stroke survivors [87]. No interventions were self-guided, suggesting that TTEAM fills a critical gap where intensive, therapist-delivered VR coordination is not necessarily needed.

Furthermore, digital technologies are increasingly highlighted as a way to scale up, and enhance accessibility and affordability of rehabilitation [88]. The World Health Organization's global strategy on digital health [89] supports this approach, and it is a key priority in the NHS Long Term Plan [32]. To this end, the UK Government has allocated £3 billion to upgrade NHS technology [33], and the 10-Year Health Plan focuses on NHS shifts from analogue to digital, hospital to community, and sickness to prevention [34]. Digital access may enable service users with less complex needs to self-manage their rehabilitation (including that related to RTW), freeing up resources for those with more complex needs, and ultimately, improving the NHS's sustainability by optimising resource allocation, reducing costs, and enhancing access to rehabilitation services.

Researchers have developed digital toolkit interventions to facilitate RTW or work retention outcomes among employees with chronic pain [90] mental health issues [46], or those recovering from gynaecological surgery [48]. A digital RTW intervention has also been developed for employers supporting employees with cancer [44]. To bring about effective, sustainable behaviour change IM recommends targeting determinants at multiple environmental levels relating to the 'at risk' population, e.g., stroke survivors, and those who can make environmental changes, e.g., employers [54]. To our knowledge, TTEAM is the only digital, self-guided RTW toolkit intervention that targets both the stroke survivor employee and their employer.

## 4.1. Theoretical contributions

Research suggests interventions based on multiple theories, targeting several constructs, have larger effects on behaviour change than those not based on theory [91]. The TTEAM co-design process was guided by IM [54], a well-established, rigorous approach for developing complex, behaviour change interventions based on multiple theories and research evidence. During IM step 2, the TDF [55] was used to identify determinants of problematic behaviours or conditions influencing employers' RTW support. Use of the TDF thus enabled us to select intervention methods that could, in theory, mitigate the negative impact of determinants. For example, stroke survivors and employers often lack understanding of the roles of RTW stakeholders who may help them to understand a stroke survivor's work capabilities, limitations, and needs. Through use of the IM taxonomy, methods such as advance organisers and active learning were selected and included within TTEAM to improve users' knowledge of stakeholders' potential roles. These methods were based on the Elaboration Likelihood Model [66,67], Social Cognitive Theory [71,72], and Theories of Information Processing [78–81]. Active learning encourages learning through goal- and activity-based experience, whilst advance organisers aid memory

formation [54,64]. Persuasive communication (based on numerous theories, see Table 4) was also used to encourage users to identify, contact, and liaise with these stakeholders to aid their learning. Collaboration across employees, employers, and healthcare professionals, for example, is crucial for facilitating shared understandings and appropriate reasonable adjustments [92,93].

## 4.2. Strengths and limitations

A strength of IM is that it encourages community-based participatory research methods to match intervention content with priority needs of the target population/s [54]. In this study, a range of relevant stakeholders, e.g., line managers, healthcare professionals, stroke survivors, business owners, HR personnel, etc, were involved in decisions on TTEAM's priority areas of focus, and on specialised content to mitigate the influence of identified determinants and problematic behaviours. For example, the needs assessment identified that stroke survivors fear highlighting (and subsequently do not disclose) limitations and needs to employers, hindering employers' support for them to RTW. The stroke survivor version of TTEAM was designed to enable and empower stroke survivors to disclose essential needs to their employer. Another issue was that employers over- or underestimated stroke survivors' work abilities, leading to too-large workloads or lack of supervisory support. The employer version was designed to enable and empower employers to increase and maintain understanding of stroke survivors' abilities, and provide reasonable adjustments when needed. If collaborative communication about an employee's support needs take place, it can lead to provision of suitable reasonable adjustments and social support, and facilitate an employee's RTW [94]. Through targeting determinants as well as behaviours, TTEAM may lead to improvements in a range of other areas, e.g., employers' competency in identifying and arranging reasonable adjustments.

Across both TTEAM versions, users are educated and equipped with tools to facilitate regular, ongoing reviews of stroke survivors' support needs and adjustments. The severity, impact and recovery trajectory of stroke can vary across individuals [95]. Ongoing RTW support tailored to a stroke survivor's needs is vital for ensuring their sustained employment. For example, even years after a stroke, new restrictions on workload and flexibility can negatively affect stroke survivors' wellbeing, and result in them leaving a work role [96]. Altogether, this highlights TTEAM's potential value as a self-guided toolkit for long-term use, ensuring sustainability of a stroke survivor's RTW. Advisory group members also believed NHS-based, therapy-led VR services may wish to signpost stroke survivor patients to TTEAM, where adequate VR is delayed or unavailable. This suggestion reaffirms TTEAM's potential to meet a critical need by improving the accessibility and scalability of VR guidance for stroke survivors and their employers.

Furthermore, advisory group members reported that they liked TTEAM's interactivity, signposting to further resources, sectioned content, ease of navigation, and multimedia, e.g., videos with accompanying text. Mayer's Cognitive Theory of Multimedia Learning suggests that messages in multimedia format, e.g., visual/pictorial and auditory/verbal are more likely to result in meaningful learning than those in one format only [97]. Group members also valued the PDF tools within TTEAM. To our knowledge, specialised sets of RTW-related tools to aid stroke survivors and employers have not yet been developed. Cognitive learning theories suggest that tools play a crucial role in facilitating learning [39,40]; they guide users through complex tasks, build deeper understanding, and develop skills and competence [98]. Furthermore, stroke survivors with various impairments were involved in pretesting and reviewing TTEAM, enabling more representative feedback regarding its accessibility, inclusivity, perceived usefulness, and ease of use/learnability. Examples of members' stroke-related impairments included fatigue, vision, pain and issues relating to communication, cognition, and sensory processing.

A limitation relates to the original focus on a toolkit for employers. Thus, stroke survivors' needs were not specifically investigated, and it is unclear *which* stroke survivors would benefit from use of TTEAM, and *when*. Involvement of employers who were also stroke survivors enabled some identification of stroke survivors' needs. Eight stroke survivors with disabilities affecting vision, stamina, language comprehension, and sensory processing, reviewed the prototypes and praised TTEAM's comprehensiveness. Refinements were suggested to mitigate technical issues and improve accessibility.

Additionally, one stroke survivor suggested that TTEAM should be offered from the point of post-stroke acute admission, and from different services, organisations, and sectors throughout their recovery journey. Further work is needed to validate the above findings.

### 4.3. Future directions

Refinement and more formal pretesting of TTEAM with larger, diverse user groups is needed. In this study, issues included PDF and narrator-speech tool incompatibility, text density, unclear navigation instructions, and lack of function to save progress. Furthermore, up to 60% of stroke survivors experience cognitive impairment post-stroke [99] which may affect their ability to recall login details, follow complex instructions, fill out forms, etc. Following initial refinements, further pretesting should include stroke survivors with cognitive impairments of varying types and severities. Observing stroke survivors' interactions with TTEAM may identify any remaining accessibility or usability issues. Qualitative data from focus groups and quantitative usability metrics, e.g., System Usability Scale scores, could enhance the validity of findings, inform decisions regarding the stroke survivor target audience, and facilitate exploration of solutions to any persistent barriers. For example, some users may prefer a printable PDF version of TTEAM, or guided support (e.g., via an embedded chatbot tool). Paper versions of TTEAM may alter the application of behaviour change methods, and diminish the potential effectiveness of TTEAM. Therefore, if implemented, these alternatives should be evaluated separately from the digital versions as comparators.

Research also indicates that stroke survivors can lack digital literacy, have limited access to technology due to economic instability or rural locations, or lack perceived benefit for engaging with digital interventions [100]. These issues may be especially prevalent among those working in industries with little need for digital skills, like unskilled manual labourers. To avoid excluding these potentially underserved populations, stroke survivor and employer representatives from various socio-economic backgrounds, occupational industries, and geographical locations should be recruited for pretesting, and during future evaluation of TTEAM. Language, cultural, religious, and ethnic differences may also hinder stroke survivors' engagement with self-management interventions. For instance, where ethnic minority communities believe only medication and rest are required for complete stroke recovery [101]. To optimise accessibility and inclusion for underserved groups, setup of the pretesting group and planning of TTEAM's implementation and evaluation (IM steps 5–6) should align with relevant equality, diversity, and inclusion guidance [102,103].

Future plans should also be informed by consultation with a core advisory group, including stroke survivors, employer and healthcare stakeholders, and professionals with expertise in implementation science, digital learning, health economics, health and safety, and equality, diversity, and inclusion. For example, further work is needed to establish *who* needs to do *what* and *when* to facilitate successful implementation (IM step 5). A detailed plan for a feasibility study with an embedded, mixed methods effectiveness-implementation hybrid design, aligned with IM step 6 and Medical Research Council guidance [104,105], is also required. This study should evaluate key feasibility outcomes such as recruitment and retention rates, acceptability, usability, intervention adherence (e.g., completion of TTEAM sections, or 'steps'), and completion of outcome measures. Progression criteria will be defined a priori, informed by literature on similar interventions and advisory group input, to determine whether a definitive trial is warranted (e.g., >70% recruitment and retention, high acceptability and usability ratings).

Potential outcomes include the TTEAM *change or performance objectives* such as employers' confidence or competence in making reasonable adjustments, *behavioural outcomes* such as employers providing reasonable adjustments in a timely way, as well as broader impacts on *health and quality-of-life improvement,* and *social* and *economic* factors, including cost-effectiveness. The use of mixed methods will also enable exploration of TTEAM's change mechanisms (i.e., the theory behind how and why it brings about intended outcomes), evaluation of its implementation and the strategies employed, and whether any refinements are needed.

Finally, other avenues of exploration may include differences in implementation and perceived applicability and relevance of TTEAM across organisational contexts. To ensure its sustainability and broader impact, potential scaling up

could also be explored, for example, developing a version for self-employed stroke survivors, producing TTEAM in alternative formats (e.g., a mobile application), or broadening its scope to employees with other health conditions or injuries attempting to RTW. Although TTEAM was developed in the UK, much of its content, particularly guidance on communication, RTW planning, and monitoring of adjustments, is likely to be transferable to settings in other countries, seeking to implement TTEAM. However, content related to UK-specific employment and disability legislation would need to be adapted to align with their national legal and policy frameworks.

## 5. Conclusion

In conclusion, TTEAM is a digital, self-guided RTW toolkit targeted at both stroke survivors and employers, addressing a critical gap in provision of VR guidance in the UK.

In this development study, use of the IM approach supported the application of theory, evidence, and stakeholder input to ensure TTEAM effectively meets users' needs when planning and managing RTW post-stroke. Further pretesting, refinement, evaluation following real-world use, and exploration of scalability are required to ensure TTEAM meets diverse user needs, and contributes long-term impact.

## Supporting information

**S1 Table. Completed GUIDED checklist.**
(PDF)

**S2 Table. Completed TIDieR checklist.**
(DOCX)

**S1 Text. Detailed description of methods.**
(DOCX)

**S3 Table. Pre-test feedback questions.**
(DOCX)

**S1 Fig. Logic model of the problem.**
(TIF)

**S2 Fig. Refined logic model of the problem.**
(ZIP)

**S4 Table. Matrices of change.**
(DOCX)

**S5 Table. Selection of behaviour change methods.**
(DOCX)

**S6 Table. Selection of practical applications.**
(DOCX)

**S7 Table. Intervention plan.**
(DOCX)

**S8 Table. Summaries of pre-test feedback.**
(DOCX)

# Author contributions

**Conceptualization:** Kristelle Craven, Jain Holmes, Jade Kettlewell, Kathryn Radford.

**Data curation:** Kristelle Craven.

**Formal analysis:** Kristelle Craven.

**Funding acquisition:** Kristelle Craven, Jain Holmes, Kathryn Radford.

**Investigation:** Kristelle Craven.

**Methodology:** Kristelle Craven.

**Project administration:** Kristelle Craven.

**Resources:** Kristelle Craven.

**Software:** Kristelle Craven.

**Supervision:** Kristelle Craven, Jain Holmes, Jade Kettlewell, Kathryn Radford.

**Validation:** Kristelle Craven.

**Visualization:** Kristelle Craven.

**Writing – original draft:** Kristelle Craven.

**Writing – review & editing:** Kristelle Craven, Jain Holmes, Jade Kettlewell, Kathryn Radford.

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
