## [Decision Letter · Decision Letter 0]

30 Apr 2025

PDIG-D-25-00219Use of intervention-mapping to co-design a self-guided, digital return-to-work toolkit for stroke survivors and employersPLOS Digital Health Dear Dr. Craven, Thank you for submitting your manuscript to PLOS Digital Health. After careful consideration, we feel that it has merit but does not fully meet PLOS Digital Health's publication criteria as it currently stands. Therefore, we invite you to submit a revised version of the manuscript that addresses the points raised during the review process. Please submit your revised manuscript within 60 days Jun 29 2025 11:59PM. If you will need more time than this to complete your revisions, please reply to this message or contact the journal office at digitalhealth@plos.org. Please include the following items when submitting your revised manuscript:* A rebuttal letter that responds to each point raised by the editor and reviewer(s). You should upload this letter as a separate file labeled 'Response to Reviewers '. This file does not need to include responses to any formatting updates and technical items listed in the 'Journal Requirements' section below.* A marked-up copy of your manuscript that highlights changes made to the original version. You should upload this as a separate file labeled 'Revised Manuscript with Track Changes '.* An unmarked version of your revised paper without tracked changes. You should upload this as a separate file labeled 'Manuscript '. If you would like to make changes to your financial disclosure, competing interests statement, or data availability statement, please make these updates within the submission form at the time of resubmission. Guidelines for resubmitting your figure files are available below the reviewer comments at the end of this letter. We look forward to receiving your revised manuscript. Kind regards, Haleh AyatollahiSection EditorPLOS Digital Health Leo Anthony CeliEditor-in-ChiefPLOS Digital Healthorcid.org/0000-0001-6712-6626 **Journal Requirements:**  1. Please amend your detailed Financial Disclosure statement. This is published with the article. It must therefore be completed in full sentences and contain the exact wording you wish to be published. i. State what role the funders took in the study. If the funders had no role in your study, please state: “The funders had no role in study design, data collection and analysis, decision to publish, or preparation of the manuscript.”  2. Please insert an Ethics Statement at the beginning of your Methods section, under a subheading 'Ethics Statement'. It must include: 1) The name(s) of the Institutional Review Board(s) or Ethics Committee(s) 2) The approval number(s), or a statement that approval was granted by the named board(s)  3) (for human participants/donors) - A statement that formal consent was obtained (must state whether verbal/written) OR the reason consent was not obtained (e.g. anonymity). NOTE: If child participants, the statement must declare that formal consent was obtained from the parent/guardian.  3. We ask that a manuscript source file is provided at Revision. Please upload your manuscript file as a .doc, .docx, .rtf or .tex.  4. Please provide separate figure files in .tif or .eps format. For more information about figure files please see our guidelines:   https://journals.plos.org/digitalhealth/s/figures https://journals.plos.org/digitalhealth/s/figures#loc-file-requirements   5. Please provide an Author Summary. This should appear in your manuscript between the Abstract (if applicable) and the Introduction, and should be 150–200 words long. The aim should be to make your findings accessible to a wide audience that includes both scientists and non-scientists. Sample summaries can be found on our website under Submission Guidelines:  https://journals.plos.org/digitalhealth/s/submission-guidelines#loc-parts-of-a-submission  6. Please ensure that all main figure files are cited in ascending numerical order in the main text of the article.  7. We have noticed that you have uploaded Supporting Information files, but you have not included a list of legends. Please add a full list of legends for your Supporting Information files after the references list.  8. In the online submission form, you indicated that “Anonymised workshop transcripts from this study may be shared for research purposes, subject to research ethics approval and a data-sharing agreement. The data custodian, Professor Kathryn Radford, can be contacted for access requests at mczkar1@exmail.nottingham.ac.uk. Additional information showing how this study was carried out, e.g., workshop PowerPoint slides, can be found at: http://doi.org/10.17639/nott.7520”. All PLOS journals now require all data underlying the findings described in their manuscript to be freely available to other researchers, either 1. In a public repository, 2. Within the manuscript itself, or 3. Uploaded as supplementary information. This policy applies to all data except where public deposition would breach compliance with the protocol approved by your research ethics board. If your data cannot be made publicly available for ethical or legal reasons (e.g., public availability would compromise patient privacy), please explain your reasons by return email and your exemption request will be escalated to the editor for approval. Your exemption request will be handled independently and will not hold up the peer review process, but will need to be resolved should your manuscript be accepted for publication. One of the Editorial team will then be in touch if there are any issues.  9. Some material included in your submission may be copyrighted. According to PLOS’s copyright policy, authors who use figures or other material (e.g., graphics, clipart, maps) from another author or copyright holder must demonstrate or obtain permission to publish this material under the Creative Commons Attribution 4.0 International (CC BY 4.0) License used by PLOS journals. Please closely review the details of PLOS’s copyright requirements here: PLOS Licenses and Copyright. If you need to request permissions from a copyright holder, you may use PLOS's Copyright Content Permission form. Please respond directly to this email or email the journal office and provide any known details concerning your material's license terms and permissions required for reuse, even if you have not yet obtained copyright permissions or are unsure of your material's copyright compatibility.  Potential Copyright Issues: a. Figure 2 contains branding/a logo. We are not permitted to publish this under our CC-BY 4.0 license, even with permission. We ask that you please remove or replace it. b. Figure 2 contains screenshots. We are not permitted to publish these under our CC-BY 4.0 license; websites are usually intellectual property and are copyrighted. This includes peripheral graphics of the web browser such as the [X] buttons. We ask that you please remove or replace it. **Additional Editor Comments (if provided):****Reviewers' Comments:** Reviewer's Responses to Questions

**Comments to the Author**

1. Does this manuscript meet PLOS Digital Health’s publication criteria ? Is the manuscript technically sound, and do the data support the conclusions? The manuscript must describe methodologically and ethically rigorous research with conclusions that are appropriately drawn based on the data presented.

Reviewer #1: Partly

Reviewer #2: Yes

2. Has the statistical analysis been performed appropriately and rigorously?

Reviewer #1: No

Reviewer #2: N/A

3. Have the authors made all data underlying the findings in their manuscript fully available (please refer to the Data Availability Statement at the start of the manuscript PDF file)?

Reviewer #1: Yes

Reviewer #2: Yes

4. Is the manuscript presented in an intelligible fashion and written in standard English?

Reviewer #1: Yes

Reviewer #2: Yes

5. Review Comments to the Author

Reviewer #1: The manuscript presents a well-structured application of intervention-mapping to co-design a digital return-to-work toolkit for stroke survivors and employers. While the study addresses an important gap in vocational rehabilitation, several revisions are needed to meet PLOS Digital Health standards and strengthen rigor:

Major Recommendations

Methodological Transparency

Clarify how workshop and advisory group participants were recruited (e.g., sampling strategy, inclusion/exclusion criteria).

Justify sample sizes for workshops (n=12 employers) and pretesting (n=15) with power calculations or saturation references.

Provide specific examples of "theory-based pretesting" methods used (e.g., cognitive walkthroughs, usability metrics).

Results Presentation

Include quantitative usability metrics (e.g., System Usability Scale scores) alongside qualitative feedback to strengthen pretesting claims.

Show comparative data on how this toolkit improves upon existing resources through tables or direct comparisons.

Ethical Compliance

Explicitly state ethics approval details (institution, reference number) in the Methods section.

Toolkit Accessibility

Address potential digital literacy barriers: 16% of UK stroke survivors have post-stroke cognitive impairments affecting technology use.

Propose alternative formats (e.g., print guides) for users with accessibility needs.

Minor Revisions

Introduction: Tighten rationale by quantifying employer knowledge gaps (e.g., "X% of UK employers report uncertainty about stroke accommodations").

Abstract: Specify the platform used (Xerte) and its accessibility features.

Non-Technical Summary: Replace subjective claims like "response was very positive" with concrete metrics (e.g., "X% of users reported improved confidence").

PLOS-Specific Requirements

Data Availability: Current statement meets requirements but should clarify timelines for data access.

Competing Interests: Adequately declared.

Financial Disclosure: Complete but should explicitly state funders' roles in study design/implementation.

Strengths to Highlight

Robust co-design process involving multiple stakeholder groups.

Clear theoretical grounding in intervention-mapping framework.

Practical focus on downloadable tools for real-world application.

This work represents a valuable contribution to digital health interventions for vocational rehabilitation. With these revisions, it will meet publication standards while maximizing impact for stroke survivors and employers navigating return-to-work challenges.

Reviewer #2: This manuscript describes the development of TTEAM, a self-guided digital return-to-work toolkit for stroke survivors and their employers. The topic is timely and addresses an important service gap. The participatory, theory-driven approach is a major strength. The paper is generally clear and thorough but could be substantially shortened and sharpened.

My only major concern is that in its present form, the study describes a development study. This is fine as I find the quality and thoroughness of presentation here to be high, however, the developmental aspect could be made more explicit as readers not familiar with all the steps of intervention-mapping may misinterpret the scope of the study and expect outcomes data. The development aspect could be made more clearer in the title, abstract, and in key paragraphs throughout the paper (e.g. last introduction paragraph and first Discussion paragraph).

Pending revision, I believe the study makes a useful contribution and will be of interest to readers of PLOS Digital Health. Below are my point by point feedback suggestions (which are generally minor in nature):

Abstract:

- Please define acronym RTW.

- Readers not familiar with intervention-mapping approach (steps 1-4), may not understand there are subsequent steps; could this be made more clear?

Introduction:

1. The Introduction/background is quite long, and could be focused better, in a manner to motivate the need and rationale behind the study more forcefully. It was hard to maintain interest in reading it due to its length and wordiness.

2. I would suggest removing tangential passages and/or making the following passages more concise:

“Authors suggested one of these interventions was effective because it encouraged women recovering from gynaecological surgery to engage in their recovery [49].”

"Stroke survivors and employers may access private VR support through their organisations (via employee assistance programs, OH provision, or group income protection insurance, for example) , or through UK government schemes such as Access to Work, or the third sector. However, support offered may fall short of meeting their needs, because it is not stroke-specific nor part of a comprehensive rehabilitative program [20]."

"In a UK survey in 2018 (N=11,134), 37% of working-age stroke survivors stopped working following stroke [27]. Specific reasons for this were not directly cited. Nevertheless, the economic, health, and social costs associated with stroke survivors’ loss of employment are vast, impacting stroke survivors, employers, and society. Reduction or loss of employment among UK stroke survivors cost £1.6 billion in 2015, and is predicted to rise 136% by 2035 [28]. Improving RTW support for the long-term sick and disabled is high priority on the UK Government’s agenda [29, 30]."

Methods:

3. Sampling: Please provide fuller demographics for participants (age, sex, time-since-stroke, etc…).

4. Framework analysis: specify coder training and any intercoder reliability process

Results:

5. Figure 1 is informative but densely formatted; consider a simplified main-text version plus a full model in Supplemental Information.

6. Figure 2 may be better suited for Supplemental Information, with each screenshot parsed out into separate figures because it is relatively dense with small font in current format.

Discussion:

7. It would be helpful to provide further details re: the planned feasibility study design (primary feasibility metrics, progression criteria, etc…).

8. It would be helpful to discuss generalizability outside the UK, especially differing employment and disability legislation.

6. PLOS authors have the option to publish the peer review history of their article (what does this mean? ). If published, this will include your full peer review and any attached files.

**Do you want your identity to be public for this peer review?** For information about this choice, including consent withdrawal, please see our Privacy Policy .

Reviewer #1: **Yes: ** Dr.Khawar Hussain Awan

Reviewer #2: No

---

## [Decision Letter · Decision Letter 1]

18 Jul 2025

Development of a digital, self-guided return-to-work toolkit for stroke survivors and employers using intervention mapping

PDIG-D-25-00219R1

Dear Dr. Craven,

We're pleased to inform you that your manuscript has been judged scientifically suitable for publication and will be formally accepted for publication once it meets all outstanding technical requirements.

Within one week, you'll receive an e-mail detailing the required amendments. When these have been addressed, you'll receive a formal acceptance letter and your manuscript will be scheduled for publication.

An invoice for payment will follow shortly after the formal acceptance. To ensure an efficient process, please log into Editorial Manager at https://www.editorialmanager.com/pdig/ click the 'Update My Information' link at the top of the page, and double check that your user information is up-to-date. For billing related questions, please contact billing support at https://plos.my.site.com/s/.

Kind regards,

Haleh Ayatollahi

Section Editor

PLOS Digital Health

Additional Editor Comments (optional):

Reviewers' comments:

Reviewer's Responses to Questions

**Comments to the Author**

1. If the authors have adequately addressed your comments raised in a previous round of review and you feel that this manuscript is now acceptable for publication, you may indicate that here to bypass the “Comments to the Author” section, enter your conflict of interest statement in the “Confidential to Editor” section, and submit your "Accept" recommendation.

Reviewer #1: All comments have been addressed

Reviewer #2: All comments have been addressed

2. Does this manuscript meet PLOS Digital Health’s publication criteria ? Is the manuscript technically sound, and do the data support the conclusions? The manuscript must describe methodologically and ethically rigorous research with conclusions that are appropriately drawn based on the data presented.

Reviewer #1: Yes

Reviewer #2: Yes

3. Has the statistical analysis been performed appropriately and rigorously?

Reviewer #1: Yes

Reviewer #2: Yes

4. Have the authors made all data underlying the findings in their manuscript fully available (please refer to the Data Availability Statement at the start of the manuscript PDF file)?

Reviewer #1: Yes

Reviewer #2: Yes

5. Is the manuscript presented in an intelligible fashion and written in standard English?

PLOS Digital Health does not copyedit accepted manuscripts, so the language in submitted articles must be clear, correct, and unambiguous. Any typographical or grammatical errors should be corrected at revision, so please note any specific errors here.

Reviewer #1: Yes

Reviewer #2: Yes

6. Review Comments to the Author

Please use the space provided to explain your answers to the questions above. You may also include additional comments for the author, including concerns about dual publication, research ethics, or publication ethics. (Please upload your review as an attachment if it exceeds 20,000 characters)

Reviewer #1: all questions has been addressed. no further explanation is required.

Reviewer #2: I deeply appreciate the Authors' thorough revision. All my concerns have been sufficiently addressed.

7. PLOS authors have the option to publish the peer review history of their article (what does this mean? ). If published, this will include your full peer review and any attached files.

**Do you want your identity to be public for this peer review?** For information about this choice, including consent withdrawal, please see our Privacy Policy . 

Reviewer #1: Yes: Dr.Khawar Hussain Awan

Reviewer #2: No
